# AKAPs-PKA disruptors increase AQP2 activity independently of vasopressin in a model of nephrogenic diabetes insipidus

Fumiaki Ando[1], Shuichi Mori[2], Naofumi Yui[1], Tetsuji Morimoto[3], Naohiro Nomura[1], Eisei Sohara[1], Tatemitsu Rai[1], Sei Sasaki[1], Yoshiaki Kondo[4], Hiroyuki Kagechika[2] & Shinichi Uchida[1]

Congenital nephrogenic diabetes insipidus (NDI) is characterized by the inability of the kidney to concentrate urine. Congenital NDI is mainly caused by loss-of-function mutations in the vasopressin type 2 receptor (V2R), leading to impaired aquaporin-2 (AQP2) water channel activity. So far, treatment options of congenital NDI either by rescuing mutant V2R with chemical chaperones or by elevating cyclic adenosine monophosphate (cAMP) levels have failed to yield effective therapies. Here we show that inhibition of A-kinase anchoring proteins (AKAPs) binding to PKA increases PKA activity and activates AQP2 channels in cortical collecting duct cells. In vivo, the low molecular weight compound 3,3′-diamino-4,4′-dihydroxydiphenylmethane (FMP-API-1) and its derivatives increase AQP2 activity to the same extent as vasopressin, and increase urine osmolality in the context of V2R inhibition. We therefore suggest that FMP-API-1 may constitute a promising lead compound for the treatment of congenital NDI caused by V2R mutations.

[1] Department of Nephrology, Tokyo Medical and Dental University (TMDU), Tokyo 113-8510, Japan. [2] Institute of Biomaterials and Bioengineering, Tokyo Medical and Dental University (TMDU), Tokyo 101-0062, Japan. [3] Division of Pediatrics, Tohoku Medical and Pharmaceutical University, Miyagi 983-8512, Japan. [4] Department of Health Care Services Management, Nihon University School of Medicine, Tokyo 173-8610, Japan. Correspondence and requests for materials should be addressed to S.U. (email: suchida.kid@tmd.ac.jp)

Congenital nephrogenic diabetes insipidus (NDI) is characterized by the inability of the kidney to concentrate urine. Daytime polyuria with nocturia and polydipsia significantly reduces the quality of life and restricts social activities of patients. In addition, persistent polyuria causes megacystis, hydroureter, hydronephrosis and other urinary tract abnormalities, leading to kidney failure[1]. Mental retardation, resulting from recurrent severe hyperosmotic dehydration and rapid over-rehydration, is critically related to prognosis[2,3]. To avoid these complications, drug discovery for congenital NDI is highly desirable.

Vasopressin type 2 receptor (V2R) and vasopressin-regulated water-channel protein aquaporin-2 (AQP2) are well-established determinants of urine concentration. In response to dehydration, the antidiuretic hormone vasopressin is secreted from the posterior pituitary. Circulating vasopressin increases water permeability of the collecting ducts by rapid translocation of AQP2 to the apical membranes, thereby inducing free water reabsorption from urine to the hypertonic interstitium to prevent further water loss. In this process, vasopressin binds to V2R, thereby activating adenylcyclase, which increases intracellular cyclic adenosine monophosphate (cAMP) production. An elevated cAMP concentration then activates cAMP-dependent protein kinase, PKA, which is thought to be involved in the phosphorylation of AQP2[4]. To date, serine 256 (S256), 261 (S261), and 269 (S269) in the C-terminus of AQP2 have been identified as major phosphorylation sites related to AQP2 trafficking[5–7]. Initial mutational analysis of AQP2 phosphorylation sites suggested that S256 was the sole phosphorylation site responsible for apical translocation of AQP2. However, the generation of phospho-specific antibodies revealed that AQP2 phosphorylation at S256 is constitutively high regardless of vasopressin stimulation[8]. On the other hand, vasopressin increases AQP2 phosphorylation at S269 and decreases AQP2 phosphorylation at S261. These changes in AQP2 phosphorylation status at S261 and S269 have been well-correlated to translocation of AQP2 to the apical plasma membrane[9,10].

Almost 80% of all congenital NDI diagnoses are caused by loss-of-function mutations of V2R[11]. Most V2R mutants are misfolded in the endoplasmic reticulum and not transported to the cell membrane[2,12]. For the treatment of congenital NDI, proper membrane sorting of V2R by restoring protein conformation or bypassing the defective V2R signaling is necessary to activate AQP2. So far, elevating cAMP levels independent of V2R has been mainly studied as a treatment option of congenital NDI. In particular, G protein-coupled receptors (GPCRs), which increase cAMP production in response to their ligands, have been intensively studied[13–17]. Nevertheless, these conventional therapeutic approaches have failed to sufficiently activate AQP2 to increase urine osmolality and no specific pharmacological drugs have yet reached clinical application.

We focused on direct activators of PKA as novel therapeutic targets of congenital NDI. PKA is known to participate in the mediation of vasopressin-induced AQP2 phosphorylation[4]. PKA is a tetramer composed of two regulatory (PKA R) and two catalytic (PKAc) subunits in its inactive form. The PKA R subunits have four isoforms: RIα, RIβ, RIIα, and RIIβ. Binding of cAMP to each PKA R subunit causes dissociation of PKAc from the PKA R subunits and subsequent phosphorylation of the consensus target sequence RRXS/T by the PKAc subunit. The intracellular distribution and substrate specificity of PKA are largely controlled by A-kinase anchoring proteins (AKAPs), which serve as scaffold proteins that tether the PKA R subunits and other signaling enzymes in proximity to their target substrates[18]. In the renal collecting ducts, AKAPs are involved in AQP2 phosphorylation[19–21]. AKAPs and PKA probably coordinate AQP2 regulation in the vasopressin signaling pathway;

however, little is known about the potential of these molecules for the treatment of congenital NDI. In this article, we report that AKAPs-PKA disruptors, which dissociate the binding of AKAPs and PKA R subunits, increased PKA activity and contributed to AQP2 phosphorylation, trafficking, and water reabsorption. The low molecular weight compound 3,3′-Diamino-4,4′-dihydroxydiphenylmethane (FMP-API-1) and its derivatives especially increased AQP2 activity to the same extent as vasopressin. AKAPs-PKA disruptors are potential novel category of therapeutic drugs for congenital NDI and other PKA-related diseases.

## Results

**Ht31 increases PKA activity in CCD cells.** The Ht31 inhibitor peptide is a stearated form of a 24-amino acid peptide derived from a human thyroid AKAP (Fig. 1a)[22,23]. This peptide competes with AKAPs for PKA RII binding and inhibits interactions between AKAPs and PKA RII. Ht31P is a control peptide with a proline substitution at the two isoleucine residues of Ht31. The role of Ht31 in PKA activity differs among target cell lines. Ht31 has been frequently used to inhibit local PKA activity by displacing PKA from its anchored sites and substrates, while Ht31 has been shown to increase PKA activity in baby hamster kidney cells[24].

Mouse principal cells of the kidney cortical collecting duct, clone 4 (mpkCCD$_{cl4}$), which exhibit endogenous expression of AQP2 and AKAPs (Supplementary Fig. 1), were used to examine the role of Ht31 in renal collecting ducts[25,26]. After the mpkCCD cells were grown to confluence on filters, Ht31 was administered to the basolateral side of the mpkCCD cells. The effect of Ht31 at 50 μM, a frequently used concentration[24,27], was compared to that of [deamino-Cys1, d-Arg8]-vasopressin (dDAVP) at 1 nM, as a positive control. PKA activity was measured by western blot analysis using a phospho-PKA substrate antibody. As shown in Fig. 1b, Ht31 and dDAVP, but not Ht31P, significantly increased PKA activity. The effects of Ht31 and dDAVP on PKA activation were inhibited by the PKA inhibitor H89 in a dose-dependent manner (Fig. 1c). Next, the intracellular levels of cAMP were measured. As expected, dDAVP significantly increased cAMP levels, while Ht31 increased PKA activity without affecting cAMP (Fig. 1d). These results demonstrated that Ht31 increased PKA activity in mpkCCD cells in the absence of vasopressin and cAMP signaling.

Dissociation of AKAP-PKA interactions by Ht31 was reported to change the subcellular localization of PKA[28]. Therefore, the effects of Ht31P and Ht31 on PKA RII localization were examined. The mpkCCD cells were either treated or not treated with Ht31 and then incubated with 1% Triton X-100 to separate the detergent-soluble cytosol fraction from the insoluble membrane fraction[29]. Although there was no change in total PKA RIIα/β protein expression, Ht31 significantly increased PKA RIIα/β protein expression in the membrane fraction (Fig. 1e, Supplementary Fig. 2). We further examined the subcellular distribution of PKA RIIα/β using immunofluorescent staining; however, the antibodies we used did not detect signals of PKA RIIα/β. Importantly, PKA activity in membrane fraction was increased after PKA was translocated from cytosol to membrane fraction (Fig. 1f). These results suggested that miss localization of intact PKA holoenzyme by Ht31 still possessed kinase activity and enhanced the phosphorylation of PKA substrates in mpkCCD cells, as described in recent reports[30].

**Ht31 activates AQP2 in CCD cells.** Next, the effects of Ht31 on AQP2 in mpkCCD cells were investigated. AQP2 phosphorylation status at S256, S261, and S269 were examined because these phosphorylation sites are responsible for AQP2 trafficking.

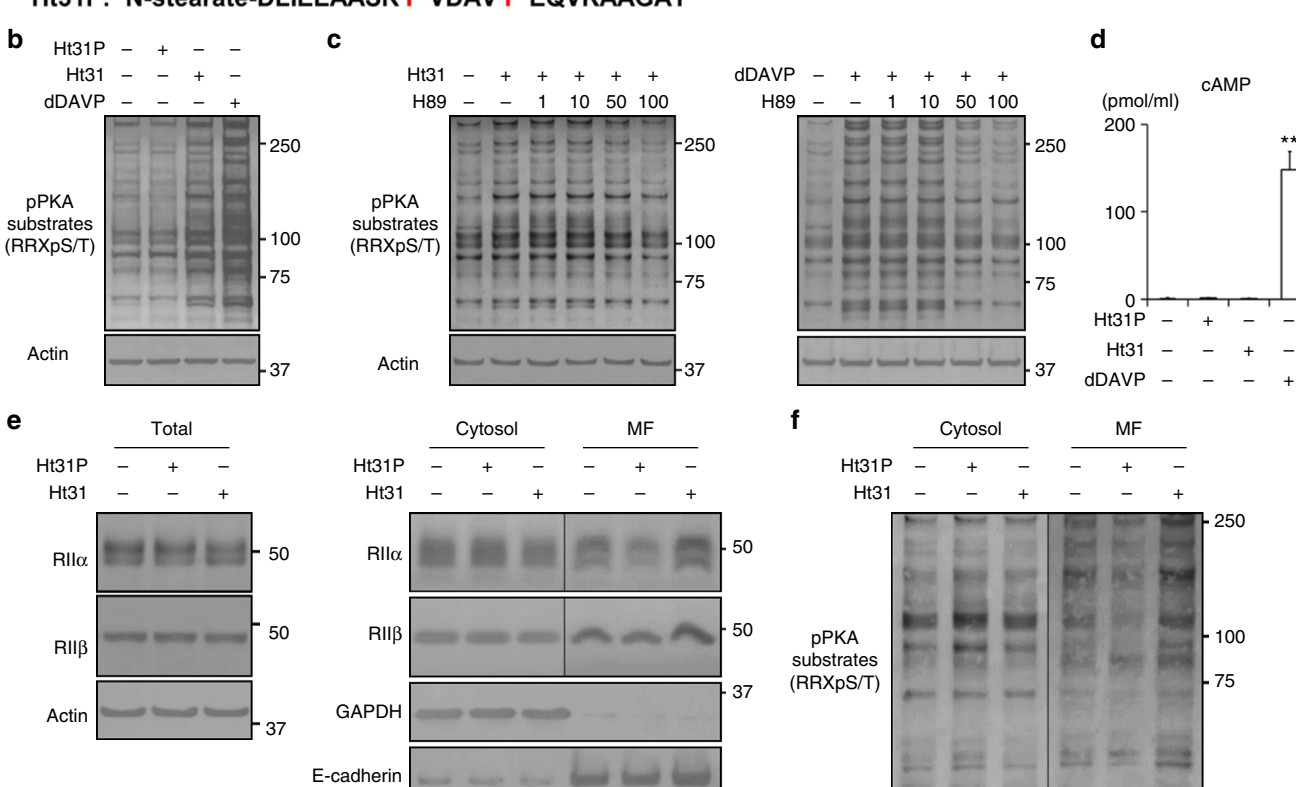

**Fig. 1** Ht31 increases PKA activity without elevation of cAMP. **a** The sequences of Ht31 and Ht31P. Ht31 and Ht31P are stearated at the N-terminus to increase cell membrane permeability. **b** The increase in PKA activity by Ht31. Ht31P (50 μM), Ht31 (50 μM), or dDAVP (1 nM) was added to the basolateral side of the mpkCCD cells for 1 h. Representative blots of three independent experiments are shown. **c** Dose-response studies of H89. Ht31 (50 μM) or dDAVP (1 nM) was added in the presence or absence of H89 (1–100 μM) for 1 h. The mpkCCD cells were pretreated with H89 for 1 h before Ht31 or dDAVP stimulation. Representative blots of three independent experiments are shown. **d** No significant elevation of cAMP concentration in response to Ht31. The mpkCCD cells were treated with Ht31P (50 μM), Ht31 (50 μM), or dDAVP (1 nM) for 1 h. Bars are mean values ± SD of three experiments. Asterisk indicates a significant difference compared with control. Tukey, **$p < 0.01$. **e**, **f** The effects of Ht31 on PKA localization and activity. Ht31P (50 μM) or Ht31 (50 μM) was added to the basolateral side of the mpkCCD cells for 1 h. The mpkCCD cells were then incubated with 1% Triton X-100 for 3 min before cell lysis and were divided into a detergent-soluble cytosol fraction and an insoluble membrane fraction (MF). Representative blots of three independent experiments are shown

Although AQP2-S256 lies within the PKA phosphorylation motif RRXS, Ht31 did not significantly phosphorylate AQP2 at S256 (Fig. 2a). Instead, Ht31 significantly phosphorylated AQP2 at S269 and significantly dephosphorylated AQP2 at S261, similar to dDAVP. The effects of Ht31 and dDAVP on AQP2 phosphorylation were inhibited by H89 (Fig. 2b, Supplementary Fig. 3), indicating that Ht31 altered AQP2 phosphorylation through PKA activation. In addition, these changes in phosphorylation status (Figs. 1b, 2a) were also inhibited by the overexpression of PKA RIIα/β (Supplementary Fig. 4). Ht31 might be mainly trapped to the overexpressed PKA RIIα/β, and could not exert the function as a AKAPs-PKA disruptor. This result ensured that the effects of Ht31 on PKA and AQP2 were mediated by PKA RIIα/β.

Because the amount of phosphorylated AQP2 at S269 is well-correlated with apical AQP2 expression in renal collecting ducts[7,31,32], the distribution of AQP2 in mpkCCD cells was examined by immunofluorescent staining. Z-stack and apical surface images demonstrated that Ht31, but not Ht31P, significantly increased apical AQP2 expression (Fig. 2c). Additionally, apical surface biotinylation analysis showed that Ht31 increased apical AQP2 expression (Fig. 2d). These results indicated that the Ht31-induced PKA activation stimulated AQP2 trafficking and apical AQP2 accumulation.

**FMP-API-1 activates PKA in CCD cells**. Ht31 effectively activated PKA and AQP2 in mpkCCD cells. Although Ht31 is a potential candidate for the treatment of congenital NDI, the in vivo efficacy of Ht31 is limited because Ht31 is a peptide drug that is characterized by a short half-life and low membrane permeability[33]. Thus, we focused on another AKAPs-PKA disruptor, FMP-API-1, which is a low molecular weight compound with enhanced plasma membrane permeability[34].

The effects of FMP-API-1 on PKA activity in mpkCCD cells was evaluated. As shown in Fig. 3a, FMP-API-1 phosphorylated PKA substrates in a dose-dependent manner, with maximum PKA activity obtained at 900 μM. The characteristics of FMP-API-1 in mpkCCD cells were extremely similar to those of Ht31. Phosphorylation of PKA substrates by FMP-API-1 were inhibited by H89 in a dose-dependent manner (Fig. 3b). In addition, FMP-API-1 increased PKA activity without cAMP elevation (Fig. 3c). Moreover, FMP-API-1 significantly increased PKA RIIα/β protein expression in the membrane fraction (Fig. 3d). In contrast to FMP-API-1, dDAVP increased the upper band of PKA RIIα in the cytosol fraction and the lower band in the membrane fraction. However, dDAVP did not induce membrane translocation of PKA RIIβ. These results indicated that FMP-API-1 activated PKA by different mechanisms from vasopressin in mpkCCD cells.

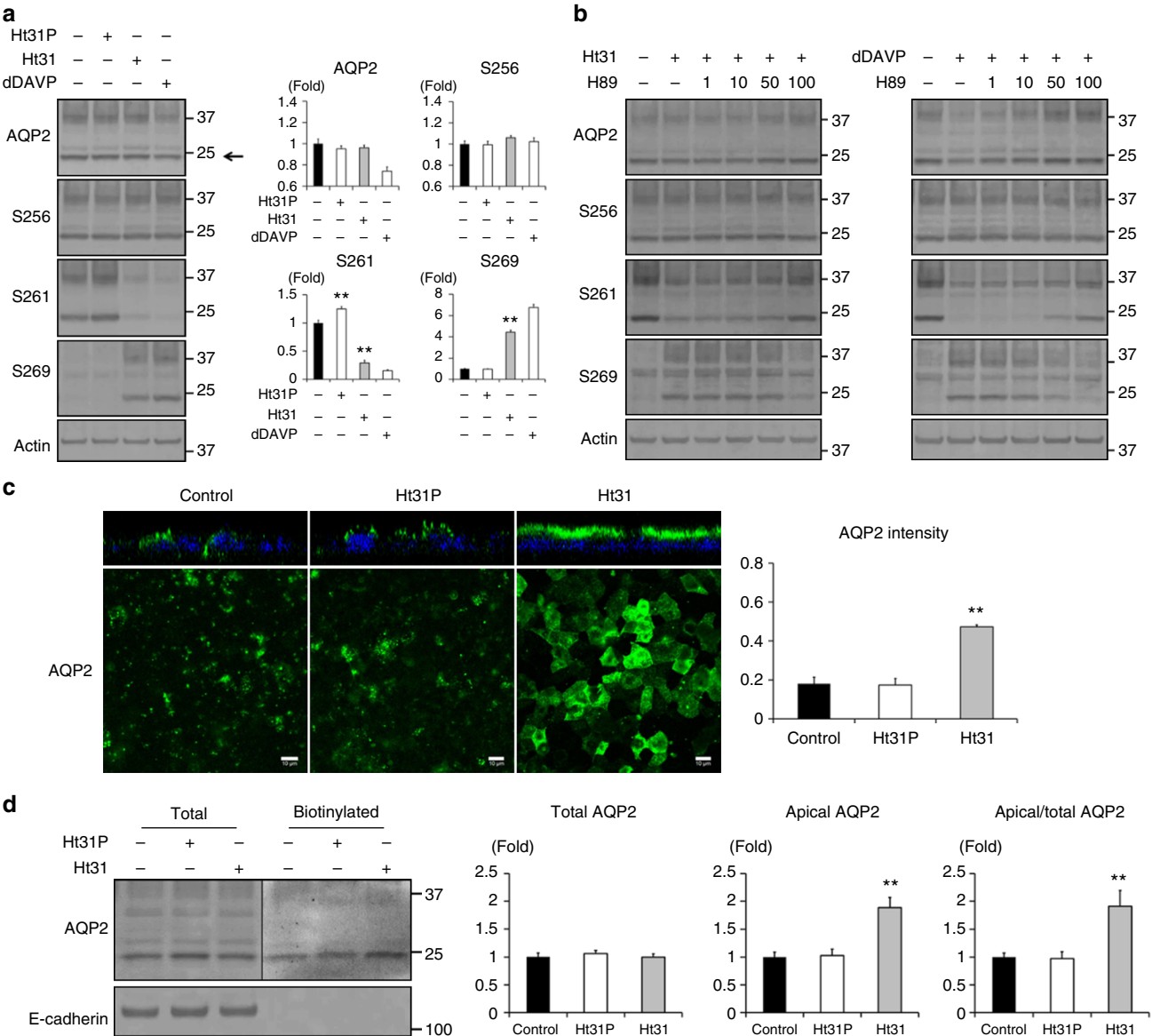

**Fig. 2** Ht31 regulates AQP2 phosphorylation and trafficking. **a** Western blot analysis of total and phosphorylated AQP2. (Left) Ht31P (50 μM), Ht31 (50 μM), or dDAVP (1 nM) was added to the basolateral side of the mpkCCD cells for 1 h. (Right) Non-glycosylated AQP2 bands (arrow) were quantified by densitometric analysis, and the results are presented in the bar graphs as the fold change, as compared to the values of the control cells. Bars are mean values ± SD of three experiments. Asterisk indicates a significant difference, as compared with the control. Tukey, **$p < 0.01$. **b** Dose-response studies of H89. Ht31 (50 μM) or dDAVP (1 nM) was added in the presence or absence of H89 (1–100 μM) for 1 h. The mpkCCD cells were pretreated with H89 for 1 h before Ht31 or dDAVP stimulation. Representative blots of three independent experiments are shown. **c** The subcellular localization of AQP2. (Left) The mpkCCD cells were treated with Ht31P (50 μM) or Ht31 (50 μM) for 1 h, and the subcellular localization of AQP2 was then analyzed by immunofluorescence using confocal microscopy. The larger panels display confocal sections of the apical regions of the cells. Z-stack confocal images are shown at the top of each panel. Representative confocal images of three independent experiments are shown. Scale bars, 10 μm. (Right) Fluorescence intensities of apical AQP2 were quantified and the results are presented in the bar graphs. Bars are mean values ± SD of three experiments. Asterisk indicates a significant difference compared with control. Tukey, **$p < 0.01$. **d** Biotinylation analysis of Ht31-induced apical AQP2 expression. (Left) The mpkCCD cells were treated with Ht31P (50 μM) or Ht31 (50 μM) for 1 h. The amount of AQP2 in the apical plasma membrane was quantitated by apical surface biotinylation. (Right) Non-glycosylated total and apical (biotinylated) AQP2 bands were quantified by densitometric analysis, and the results are presented in the bar graphs as the fold change, as compared to the values of the control cells. Bars are mean values ± SD of three experiments. Asterisk indicates a significant difference, as compared with the control. Tukey, **$p < 0.01$

**FMP-API-1 increases AQP2 activity in vitro**. As Ht31-induced AQP2 trafficking (Fig. 2c, d), PKA activation by FMP-API-1 also increased AQP2 activity in mpkCCD cells. FMP-API-1 significantly phosphorylated AQP2 at S269 and significantly dephosphorylated AQP2 at S261, while AQP2 phosphorylation at S256 was constitutively high (Fig. 4a). These changes in AQP2 phosphorylation status were inhibited by H89 (Fig. 4b,

Supplementary Fig. 5a). Immunofluorescent analysis showed that FMP-API-1 significantly increased apical AQP2 expression (Fig. 4c). Remarkably, the efficacy of FMP-API-1 in AQP2 trafficking was almost equal to that of dDAVP. Also, apical surface biotinylation analysis showed that FMP-API-1 increased apical AQP2 expression to the same extent as dDAVP (Fig. 4d, Supplementary Fig. 5b). Functional analysis of osmotic water

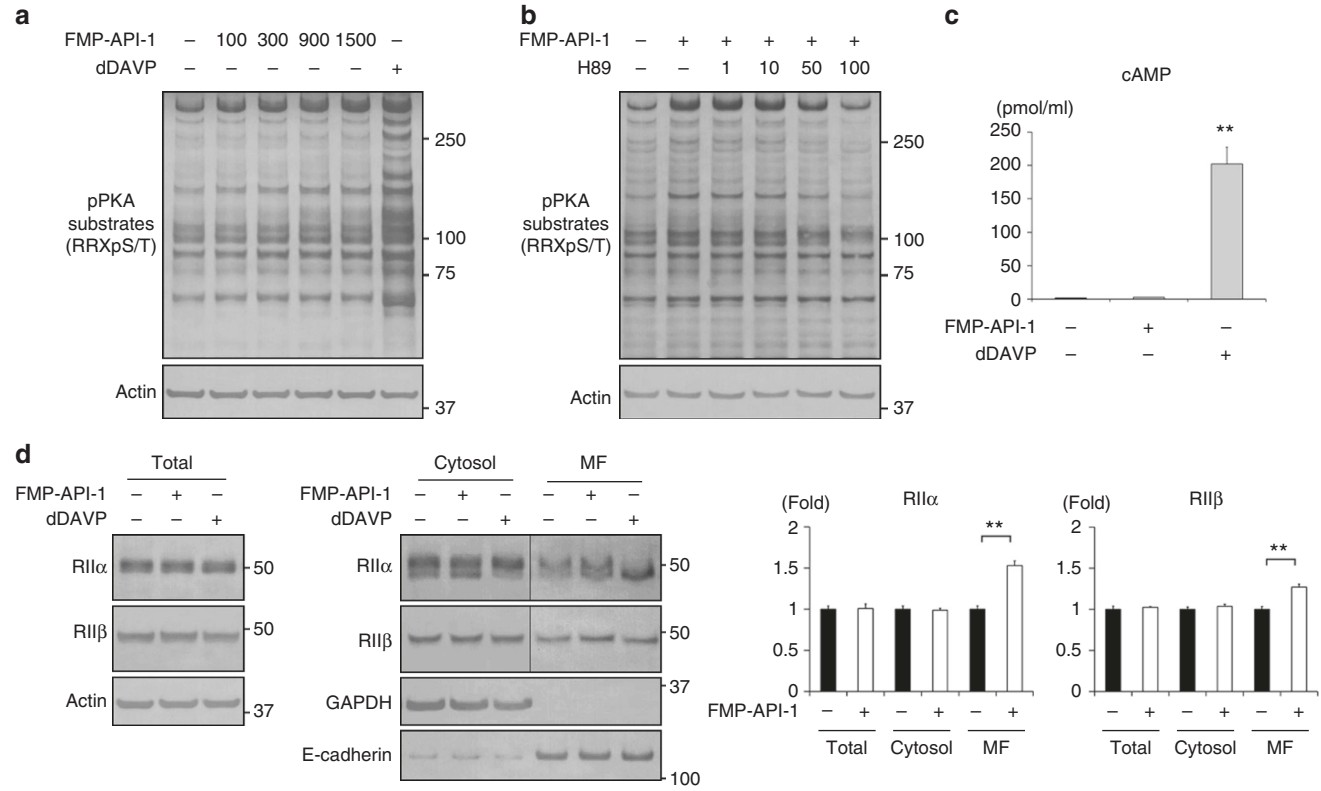

**Fig. 3** FMP-API-1 increases PKA activity without elevation of cAMP. **a** Dose-response studies of FMP-API-1. FMP-API-1 (100–1500 μM) was added to the basolateral side of the mpkCCD cells for 1 h. Representative blots of three independent experiments are shown. **b** Dose-response studies of H89. FMP-API-1 (900 μM) was added in the presence or absence of H89 (1–100 μM) for 1 h. The mpkCCD cells were pretreated with H89 for 1 h before FMP-API-1 stimulation. Representative blots of three independent experiments are shown. **c** No significant elevation of cAMP concentration in response to FMP-API-1. The mpkCCD cells were treated with FMP-API-1 (900 μM) for 1 h. Bars are mean values ± SD of three experiments. Asterisk indicates a significant difference, as compared with the control. Tukey, $**p < 0.01$. **d** The effects of FMP-API-1 on PKA localization. (Left) FMP-API-1 (900 μM) was added to the basolateral side of the mpkCCD cells for 1 h. The mpkCCD cells were then incubated with 1% Triton X-100 for 3 min before cell lysis and divided into a detergent-soluble cytosol fraction and an insoluble membrane fraction. Representative blots of three independent experiments are shown. (Right) Bands of PKA RIIα/β were quantified by densitometric analysis, and the results are presented in the bar graphs as the fold change, as compared with the values of the control cells. Bars are mean values ± SD of three experiments. Tukey, $**p < 0.01$

permeability ($P_f$) in isolated tubules of the CCDs of mouse kidneys was further examined using a microperfusion technique. As shown in Fig. 4e, FMP-API-1 significantly increased $P_f$, as compared to the control values. The effect of FMP-API-1 was equivalent to that of dDAVP. These data indicated that FMP-API-1 certainly increased AQP2 activity and contributed to transcellular water transport across the tubular epithelial cells.

**FMP-API-1 increases AQP2 activity in an NDI mouse model.** Considering that FMP-API-1 activated AQP2 without elevating cAMP, FMP-API-1 appears to be a promising therapeutic target for congenital NDI caused by V2R mutations. Therefore, the therapeutic effects of FMP-API-1 in a V2R-inhibited NDI mouse model were examined. At first, C57BL/6 mice were housed individually in metabolic cages for 24 h to measure basal urine osmolality, urine output, and water intake. The C57BL/6 mice were then subcutaneously infused with tolvaptan or tolvaptan plus FMP-API-1 using osmotic minipumps[35] and were housed again in metabolic cages for another 24 h to measure water balance. As shown in Fig. 4f, tolvaptan decreased urine osmolality and increased urine output and water intake. Administration of FMP-API-1 significantly attenuated the effects of tolvaptan. FMP-API-1 successfully increased the urine concentrating ability in the NDI mouse model.

**The derivatives of FMP-API-1 robustly activates AQP2.** FMP-API-1 is a novel lead compound that can strongly activate AQP2. We next synthesized derivatives of FMP-API-1 to obtain greater pharmacological potency. As shown in Fig. 5a, the aromatic residues of FMP-API-1/27 and FMP-API-1/28 contain hydroxyl (–OH), but not amino (–NH$_2$), groups that can inhibit the binding of AKAPs and PKA R subunits more efficiently than FMP-API-1[34], and have more potential to activate AQP2. In the mpkCCD cells, FMP-API-1/27 indeed activated PKA and AQP2 phosphorylation at lower concentrations, as compared with FMP-API-1 (Fig. 5b, Supplementary Fig. 6a). However, the effects of FMP-API-1/28 on PKA and AQP2 phosphorylation remained the same as those of FMP-API-1 (Supplementary Fig. 7).

Based on these results, FMP-API-1/27 was expected to exert greater in vivo efficacy than FMP-API-1. Thus, wild-type mice were intraperitoneally injected with FMP-API-1/27 to evaluate AQP2 phosphorylation. Surprisingly, FMP-API-1/27 had an unprecedented high potency and strongly changed AQP2 phosphorylation at S261 and S269 to the same extent as dDAVP (Fig. 5c, Supplementary Fig. 6b). Although AQP2-S256 is highly phosphorylated at baseline in vivo kidney[8], FMP-API-1/27 further increased AQP2 phosphorylation at S256. Hence, FMP-API-1/27 possesses unique properties in the regulation of AQP2 phosphorylation.

We next examined the effects of FMP-API-1/27 in the NDI mice model. In brief, C57BL/6 mice were subcutaneously

infused with tolvaptan for 24 h and then intraperitoneally injected with FMP-API-1/27. As shown in Fig. 5d, tolvaptan significantly suppressed AQP2 phosphorylation at S256 (Supplementary Fig. 6c)[35]. On the contrary, FMP-API-1/27 clearly counteracted the effects of tolvaptan and activated AQP2 phosphorylation. FMP-API-1/27-induced phosphorylation of AQP2 was sufficient to increase urine concentration ability. Tolvaptan significantly decreased urine osmolality, and FMP-API-1/27 significantly attenuated the effects of tolvaptan (Fig. 5e).

FMP-API-1/27 was also effective via an alternative route of administration. After intraperitoneal injection of FMP-API-1/27, AQP2 phosphorylation was observed for up to 3 h, whereas subcutaneous injection of FMP-API-1/27 phosphorylated AQP2 at S256 and S269 for more than 6 h (Fig. 5f, Supplementary Fig. 6d, e). Hence, the long-term effects of FMP-API-1/27 may reduce nocturia frequency in patients with congenital NDI by self-injection of FMP-API-1/27 before bedtime.

In addition to potency, drug specificity is an important pharmaceutical property. Although PKA is expressed

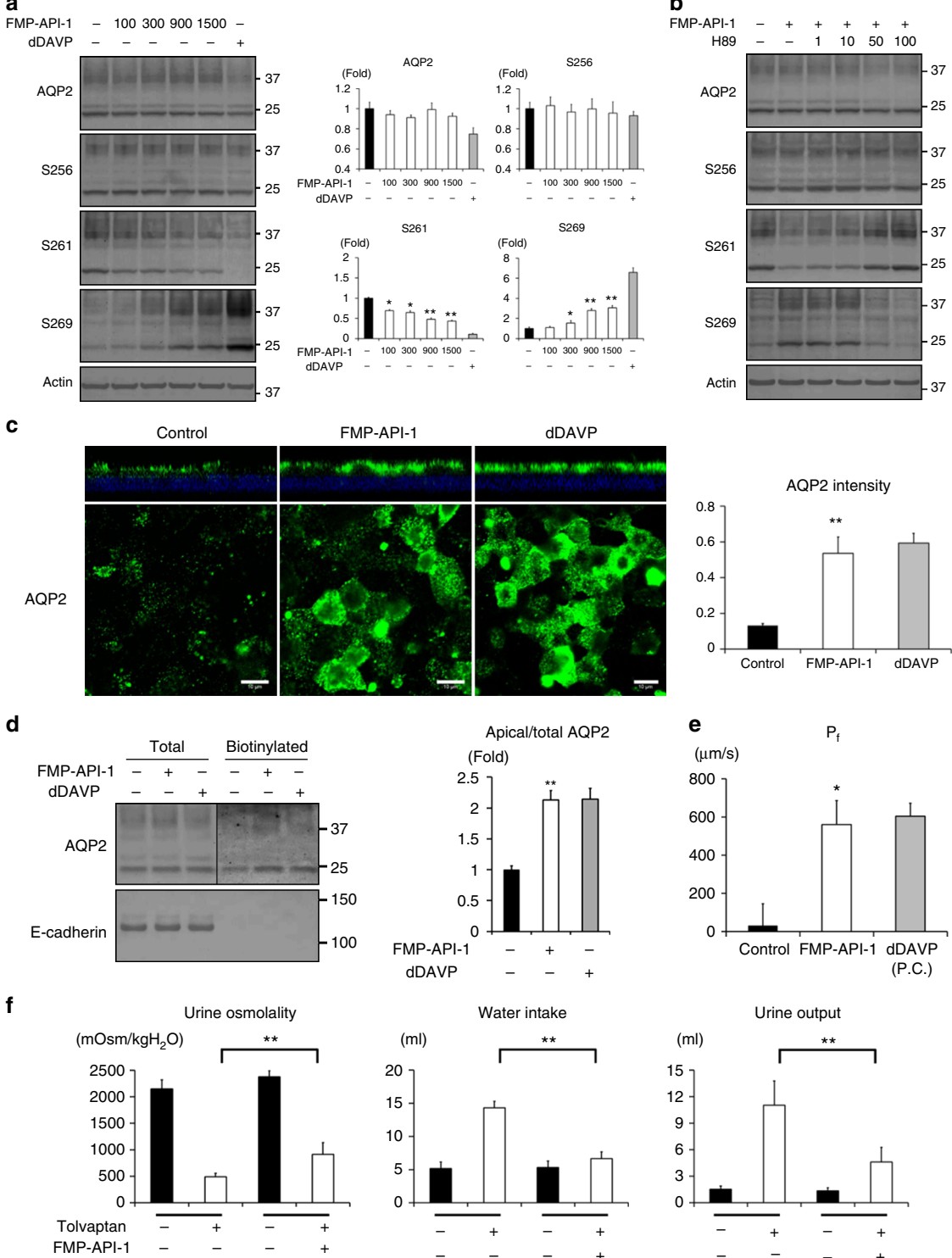

ubiquitously, FMP-API-1/27 did not phosphorylate all of the PKA substrates. After FMP-API-1/27 administration, phosphorylation of PKA substrates in the whole kidney and heart were not significantly increased (Fig. 5g). Moreover, a representative substrate of PKA, phospho cardiac troponin I (p-cTnI), also did not respond to FMP-API-1/27. These results indicated that FMP-API-1/27-induced phosphorylation of AQP2 was tissue-specific.

## Discussion

The results of the present study clarified that FMP-API-1 and its derivatives robustly activated AQP2. In renal collecting ducts, dissociation of AKAP binding to the PKA R subunits was found to contribute to activation of PKA and AQP2. The efficacy of FMP-API-1 and FMP-API-1/27 in terms of AQP2 phosphorylation, trafficking, and osmotic water transport was almost equal or even superior to that of an endogenous activator of vasopressin (Fig. 4a-e, Fig. 5b, c), which has never been accomplished before. Moreover, FMP-API-1 and FMP-API-1/27 also possessed high plasma membrane permeability, and indeed reduced urine output in the NDI mouse model (Figs. 4f, 5d, e, 6).

FMP-API-1 exhibited not only high potency to AQP2, but also enhanced substrate specificity. In the mpkCCD cells, FMP-API-1 phosphorylated a specific subset of target proteins (Fig. 3b, lanes 1 and 2). Similarly, only a part of the PKA substrates were significantly phosphorylated by FMP-API-1 in cardiac myocytes[34]. In contrast, GPCR agonists, such as vasopressin and isoproterenol, phosphorylated almost all the PKA substrates in the target cells by elevating cAMP levels (Fig. 3a, c)[34]. The direct PKA activator FMP-API-1 and the cAMP/PKA activators differentially regulated phosphorylation of PKA substrates. Another important finding was that FMP-API-1/27 demonstrated tissue-specific efficacy in vivo. FMP-API-1/27 exclusively increased PKA activity in renal collecting ducts without phosphorylation of most of the PKA substrates in the whole kidney and heart (Fig. 5g). We further confirmed that the representative cardiac PKA substrate p-cTnI did not respond to FMP-API-1/27. The target-selective property of FMP-API-1/27 would be advantageous to reduce the risk of adverse drug events because PKA is expressed ubiquitously and regulates many substrates.

FMP-API-1/27 highly activated AQP2 phosphorylation in vivo, rather than in vitro (Fig. 5b, c), which is a more favorable feature for the treatment of congenital NDI. The increase in AQP2 phosphorylation at S256 was only detected in the mouse kidney. These differences were probably caused by the dissimilar expression patterns of AKAPs and PKA R subunits among the target cells. To date, 43 genes and more than 70 functionally distinct AKAP proteins have been identified[36]. While the

majority of AKAPs preferentially bind to PKA RII, others have PKA RI specificity or can bind to both PKA RI and PKA RII[37,38]. Various combinations and interactions of AKAPs and PKA R subunits are, respectively, responsible for the potency and specificity of PKA activity[39]. Although the precise target of FMP-API-1/27 in the collecting ducts remains unknown, the results of this study demonstrated that the disruption of AKAPs and PKA R subunits in the mouse kidney clearly promoted AQP2 phosphorylation. Interestingly, the expression patterns of AKAPs and PKA R subunits between the CCD and medullary collecting duct (MCD) were quite different. RNA-seq database analysis revealed that *Akap11*(AKAP220), *Akap12*, *Cbfa2t3*, and *Prkar2b* (PKA RIIβ) were mostly expressed by the MCD, but not the CCD[40]. AKAP220 actually colocalizes with AQP2 in the MCD, where it is involved in AQP2 phosphorylation[21]. These different gene expression profiles between the CCD and MCD cells may explain the more potent effect of FMP-API-1/27 in the mouse kidney, as compared with its effect in mpkCCD cells, a CCD cell line. Moreover, these different expression patterns of AKAPs and PKA in each tissue may contribute to the target-selective property of FMP-API-1/27. The development of other AKAPs-PKA disruptors targeting novel combinations of AKAPs and PKA R subunits is a potential therapeutic strategy for various PKA-related diseases. However, the question of whether AKAPs-PKA disruptors could be used in clinical settings is still controversial, and remains to be addressed. In particular, the precise mechanisms underlying the efficacy of such small molecule inhibitors, including FMP-API-1/27, still need to be elucidated. In addition, FMP-API-1/27 showed little effect in mice by oral administration (F.A., unpublished observations), suggesting that further modification of the compound would be necessary to improve its pharmacokinetics.

The role of PKA in the regulation of AQP2 phosphorylation remains to be elucidated. Generally, PKA is considered to be responsible for AQP2 phosphorylation at S256[4,5] because vasopressin increases cAMP/PKA activity and AQP2-S256 is within the PKA targeting motif (RRXS/T). In vitro kinase analysis supports this notion. Purified PKA significantly phosphorylated AQP2 peptide at S256[7]. Conversely, PKA failed to regulate the AQP2 peptide at S261 and S269[7,41]. These results suggest that vasopressin-induced PKA activation selectively increases AQP2 phosphorylation at S256. However, in the mouse kidney, vasopressin regulates AQP2 phosphorylation at S261 and S269, rather than S256[8], suggesting that unidentified kinases or phosphatases, other than PKA, are present, in order to account for the detailed mechanisms of AQP2 phosphorylation in the vasopressin signaling pathway[42]. Importantly, the direct PKA activator FMP-API-1/27 robustly

**Fig. 4** FMP-API-1 increases urine concentrating ability. **a** Dose-response studies of FMP-API-1. (Left) FMP-API-1 (100–1500 μM) was added to the basolateral side of the mpkCCD cells for 1 h. (Right) Non-glycosylated AQP2 bands were quantified by densitometric analysis, and the results are presented in the bar graphs as the fold change, as compared to the values of the control cells. Bars are mean values ± SD of three experiments. Asterisk indicates a significant difference compared with control. Tukey, *$p < 0.05$, **$p < 0.01$. **b** Dose-response studies of H89. FMP-API-1 (900 μM) was added in the presence or absence of H89 (1–100 μM) for 1 h. The mpkCCD cells were pretreated with H89 for 1 h before FMP-API-1 stimulation. Representative blots of three independent experiments are shown. **c** Immunofluorescent analysis of FMP-API-1-induced apical AQP2 expression. (Left) The mpkCCD cells were treated with FMP-API-1 (900 μM) or dDAVP (1 nM) for 1 h. Immunofluorescence staining of AQP2 was analyzed as in Fig. 2c. Scale bars, 10 μm. (Right) Bars are mean values ± SD of three experiments. Asterisk indicates a significant difference compared with control. Tukey, **$p < 0.01$. **d** Biotinylation analysis of FMP-API-1-induced apical AQP2 expression. (Left) The mpkCCD cells were treated with FMP-API-1 (900 μM) for 1 h. The amount of AQP2 in the apical plasma membrane was analyzed as in Fig. 2d. (Right) Bars are mean values ± SD of three experiments. Asterisk indicates a significant difference compared with control. Tukey, **$p < 0.01$. **e** $P_f$ in the mouse CCD. FMP-API-1 (900 μM) was added to isolated CCD tubules for 1 h. After FMP-API-1 washout, dDAVP (1 nM) was added to isolated CCD tubules for 15 min. Each value is an average of triplicate assays. P.C. indicates positive control. Mean values ± SE were determined from five experiments. Student's *t*-test, *$p < 0.05$. **f** Increase in urine concentrating ability by FMP-API-1 in the tolvaptan-infused mice. After measurement of basal water balance in metabolic cages for 24 h, C57BL/6 mice were subcutaneously infused with tolvaptan (25 mg/kg/day) or tolvaptan plus FMP-API-1 (25 mg/kg/day) for 24 h using osmotic minipumps. The results are presented in the bar graphs. Bars are mean values ± SD from five or six experiments. Tukey, **$p < 0.01$

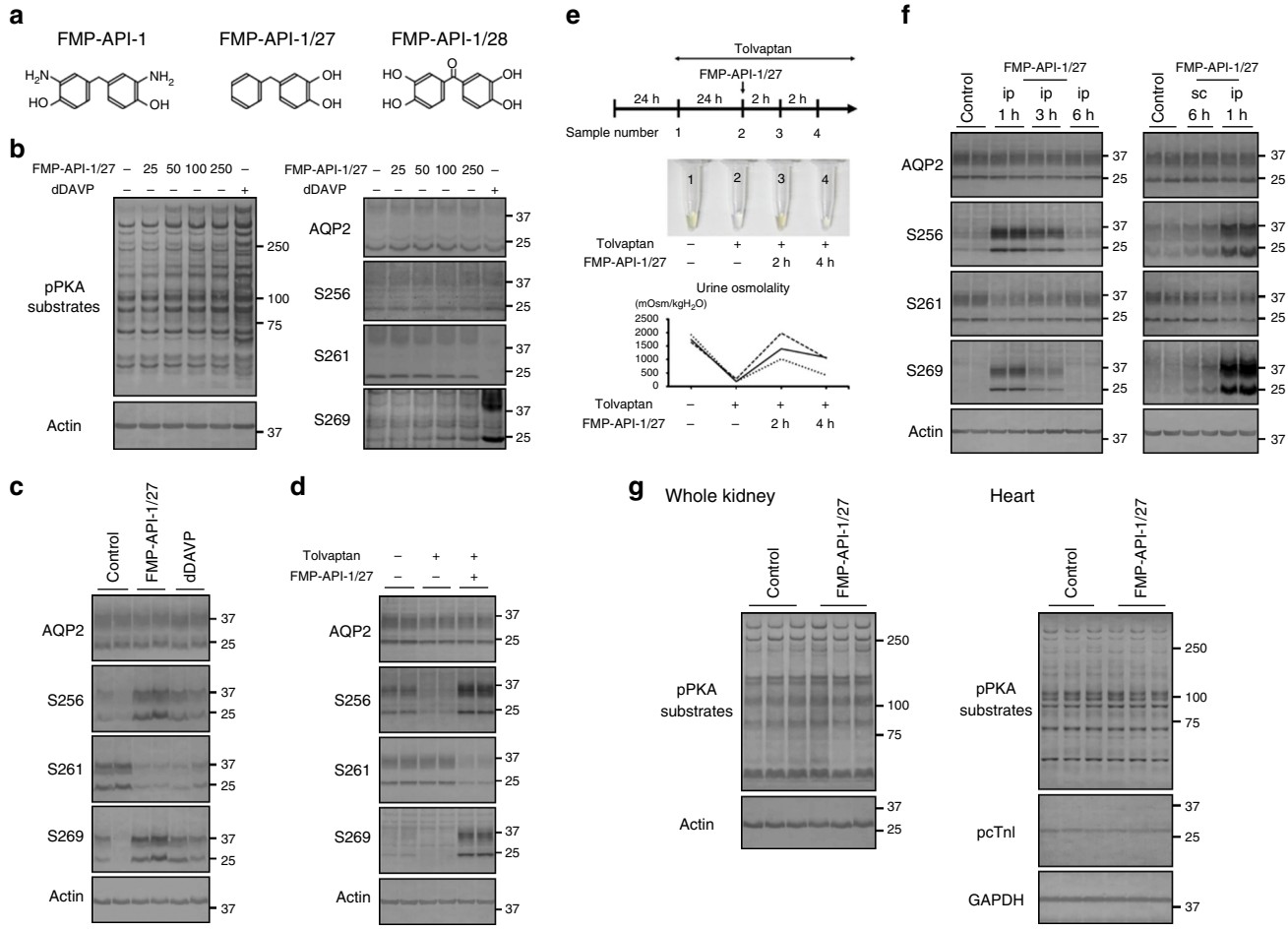

**Fig. 5** Derivatives of FMP-API-1 robustly activate AQP2. **a** Derivatives of FMP-API-1. The chemical formulas of FMP-API-1, FMP-API-1/27, and FMP-API-1/28 are shown. **b** Dose-response studies of FMP-API-1/27. FMP-API-1/27 (25–250 μM) was added to the basolateral side of the mpkCCD cells for 1 h. Representative blots of three independent experiments are shown. **c** Western blot analysis of AQP2 in mouse kidneys. The C57BL/6 mice were intraperitoneally infused with DMSO, FMP-API-1/27 (30 mg/kg), or dDAVP (100 μg/kg). Samples were collected 1 h after intraperitoneal infusion. Representative blots of four independent experiments are shown. **d** AQP2 phosphorylation by FMP-API-1/27 in the NDI mouse model. C57BL/6 mice were subcutaneously infused with DMSO or tolvaptan (25 mg/kg/day) for 24 h using osmotic minipumps, and then intraperitoneally infused with DMSO or FMP-API-1/27 (30 mg/kg). Samples were collected 1 h after the intraperitoneal infusion. Representative blots of four independent experiments are shown. **e** Increase in urine concentrating ability by FMP-API-1/27 in the NDI mouse model. (Upper) The time course of the experiment is shown. After measurement of basal urine osmolality in metabolic cages for 24 h, C57BL/6 mice were subcutaneously infused with tolvaptan (25 mg/kg/day) for 24 h using osmotic minipumps. C57BL/6 mice were then intraperitoneally infused with FMP-API-1/27 (30 mg/kg). Urine samples were collected at the indicated times. (Middle) Representative urine samples of three independent experiments are shown. (Lower) The time course of urine osmolality is shown. **f** Drug effects of FMP-API-1/27 via the different routes of administration. The C57BL/6 mice were treated with intraperitoneal (ip; 30 mg/kg) or subcutaneous (sc; 80 mg/kg) injection of FMP-API-1/27. Samples were collected at the indicated times. Representative blots of four independent experiments are shown. **g** The effects of FMP-API-1/27 on PKA activity. The C57BL/6 mice were intraperitoneally infused with DMSO or FMP-API-1/27 (30 mg/kg). Samples were collected 1 h after intraperitoneal infusion. Representative blots of three independent experiments are shown

increased AQP2 phosphorylation at S256 in the mouse kidney, which has never been observed before (Fig. 5c). FMP-API-1/27 probably phosphorylated AQP2-S256 directly through PKA activation and also activated its downstream signaling pathway, thereby inducing AQP2 apical translocation with changes to AQP2-S261 and AQP2-S269 phosphorylation. These results clarified that PKA certainly possessed AQP2-activating effects in renal collecting ducts, indicating that direct PKA activators should be further investigated as novel therapeutic targets of congenital NDI.

In conclusion, FMP-API-1 was identified as a reliable lead compound for the treatment of congenital NDI. The development of FMP-API-1 derivatives presents a novel therapeutic strategy. In addition to potency and specificity, subcutaneous administration of FMP-API-1/27 had long-term effects on AQP2

(Fig. 5f). FMP-API-1/27 and the derivatives of FMP-API-1 are promising drug candidates to improve the quality of life of patients with congenital NDI.

## Methods

**Cell culture**. mpkCCD_{cl4} cells (gift from Alain Vandewalle, Paris) were cultured as previously described[25]. The mpkCCD cells were seeded on semipermeable filters (Transwell, 0.4-μm pore size; Corning Costar). For immunofluorescence and western blot analysis, 0.33-cm² and 4.67-cm² filters were used, respectively. The mpkCCD cells were cultured for 5 days, with changes of the medium daily. St-Ht31 (Promega Corporation, Madison, WI, USA), St-Ht31P (Promega Corporation), FMP-API-1 (Sigma–Aldrich Corporation, St. Louis, MO, USA), FMP-API-1/27, FMP-API-1/28, dDAVP (Sigma–Aldrich Corporation), and H89 (Sigma–Aldrich Corporation) were applied to the basolateral side of the mpkCCD cells.

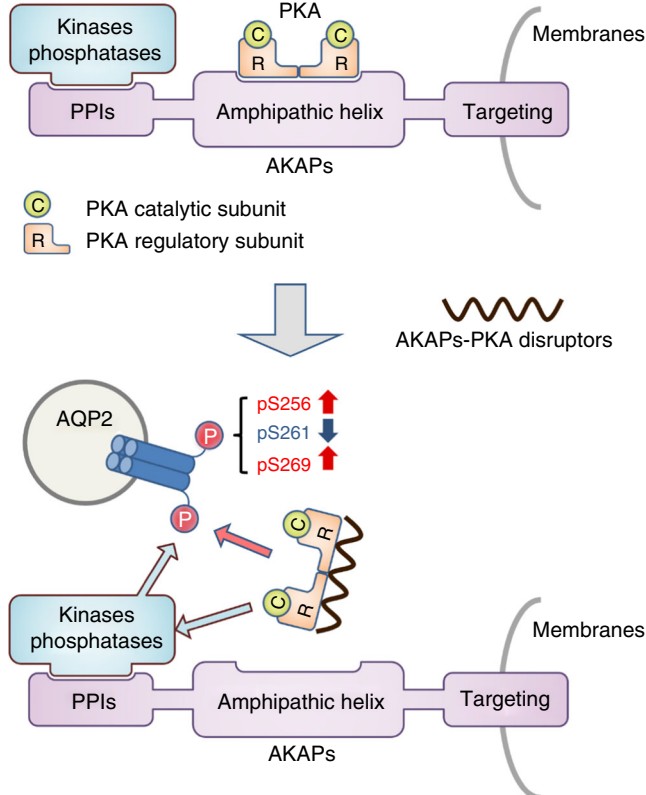

**Fig. 6** Schematic summary of the effects of FMP-API-1 and its derivatives on AQP2. AKAPs generally contain protein-protein interactions (PPIs) sites, amphipathic helix domain, and targeting domain[48]. The amphipathic α-helix enables the anchoring of PKA holoenzymes. The targeting domain mediates the tethering of PKA complexes to selected subcellular compartments. FMP-API-1 and FMP-API-1-1/27 dissociated PKA from AKAPs. The changes in PKA localization increased PKA activity in renal collecting ducts. PKA as well as additional kinases and phosphatases which are interacted with AKAPs through PPIs sites coordinately regulated AQP2 phosphorylation, trafficking, and water transport. In contrast to vasopressin, FMP-API-1/27 increased phosphorylation of AQP2 at S256 in the mouse kidney. FMP-API-1 and FMP-API-1/27 successfully improved urine concentrating ability in the NDI mouse model

**Animal experiments**. The animal studies were performed using 10-week-old male C57BL/6 mice that were housed individually in metabolic cages with free access to food and water to measure urine osmolality, urine output, and water intake. The mice were subcutaneously infused with tolvaptan (LKT Laboratories, Inc., St. Paul, MN, USA) or tolvaptan plus FMP-API-1 using osmotic minipumps (Alzet model 1003D; ALZA Corporation, Cupertino, CA, USA) as previously described[35]. Urine osmolality was measured with a Fiske One-ten Osmometer (John Morris Scientific Pty Ltd., Chatswood, NSW, Australia). The protocols of the animal experiments were approved by the Animal Care and Use Committee of Tokyo Medical and Dental University.

**Western blot analysis**. The mpkCCD cells were washed twice with phosphate-buffered saline (PBS) and then solubilized in 200 μL of lysis buffer as previously described[32]. For analysis of the cell fractions, the mpkCCD cells were incubated with 650 μL of 1% Triton X-100 in PBS for 3 min before cell lysis, yielding a detergent-soluble cytosol fraction and an insoluble membrane fraction[29]. After centrifugation at $15,000 \times g$ for 10 min at 4 °C, the protein concentration was measured using the Bradford protein assay method (Expedeon Inc., San Diego, CA, USA). The supernatants were denatured in sodium dodecyl sulfate (SDS) sample buffer (Cosmo Bio, Co., Ltd., Tokyo Japan) for 30 min at 37 °C. Equal amounts of protein were separated by SDS-polyacrylamide gel electrophoresis and then transferred to nitrocellulose membranes (GE Healthcare Life Sciences, Chicago, IL, USA). The blots were probed with the following primary antibodies: goat anti-AQP2 (dilution, 1:1000; N-20, sc-9880; Santa Cruz Biotechnology, Inc., Dallas, TX, USA), rabbit anti-AQP2 (dilution, 1: 1000; phospho S256, ab109926; Abcam, Cambridge, UK), rabbit anti-AQP2 (dilution, 1: 1000; phospho S261, p112-261; Symansis, Levels, New Zealand), rabbit anti-AQP2 (dilution, 1:1000; phospho S269,

p112-269; Symansis), mouse anti-β-actin (dilution, 1:1000; A2228; Sigma–Aldrich Corporation), rabbit anti-phospho-PKA substrate (dilution, 1:1000; #9624; Cell Signaling Technology, Inc., Beverly, MA, USA), mouse anti-PKA RIIα (dilution, 1:1000; #612243; BD Transduction Laboratories, San Jose, CA, USA), mouse anti-PKA RIIβ (dilution, 1:1000; #610626; BD Transduction Laboratories), mouse anti-glyceraldehyde 3-phosphate dehydrogenase (dilution, 1:1000; sc-32233; Santa Cruz Biotechnology, Inc.), and mouse anti-E-cadherin (dilution, 1:1000; #610182; BD Transduction Laboratories). Alkaline phosphatase-conjugated anti-IgG antibody (Promega Corporation) was used as the secondary antibody and Western Blue (Promega Corporation) was used to detect the signals. The band intensities of the western blots were quantified using ImageJ software (https://imagej.nih.gov/ij/). Uncropped western blots are shown in Supplementary Figure 8–14.

**Immunofluorescence studies**. The mpkCCD cells were fixed with 4% paraformaldehyde and permeabilized with 0.1% Triton-X/PBS. The filters were detached from the holders and incubated with goat anti-AQP2 antibody (N-20; dilution, 1:500). Alexa 488 dye-labeled antibodies (dilution, 1:200; Molecular Probes, Carlsbad, CA, USA) were used as the secondary antibodies. The samples were mounted with Vectashield/4′,6-diamidino-2-phenylindole (Vector Laboratories, Burlingame, CA, USA). Immunofluorescence images were obtained using the Zeiss LSM 510 Meta Confocal Microscope (Carl Zeiss AG, Oberkochen, Germany). The fluorescence intensities of AQP2 in the mpkCCD cells were quantified using ImageJ software.

**Reverse transcription-PCR analysis**. Total RNA from the mpkCCD cells and C57BL/6 mouse was extracted using TRIzol Reagent (Invitrogen, Carlsbad, CA, USA) and purified with RNase-free DNase Sets and RNeasy Kits (Qiagen, Venlo, Netherlands). RNAs were reverse-transcribed using Omniscript Reverse Transcriptase (Qiagen). Primer sequences used for PCR were summarized in Supplementary Table 1.

**Lentiviral vector production and transduction**. The mouse *Prkar2a* and *Prkar2b* coding regions were amplified from mpkCCD-derived cDNA by PCR and were cloned into pENTRegfp2 (Addgene, Cambridge, MA, USA #22450) using the Gibson assembly technique (New England Biolabs, Ipswich, MA, USA). C-terminal GFP-tagged PKA RIIα and PKA RIIβ were subsequently transferred into pLenti CMV Puro DEST (Addgene #17452) using the LR reaction. The recombinant lentiviral vectors, packaging vector psPAX2 (Addgene #12260) and enveloping vector pMD2.G (Addgene #12259) were co-transfected into HEK 293T cells. The culture medium was harvested and centrifuged at $500 \times g$ for 5 min at 4 °C to remove any cellular debris. The supernatant containing 4 μg/ml polybrene (Sigma–Aldrich Corporation) was added to the mpkCCD cells. The cells were centrifuged at $1,000 \times g$ for 90 min at room temperature to increase transduction efficiency. Successful transduction was confirmed by GFP positive cells under a fluorescence microscope. For molecular analysis mixed populations of cells without antibiotic selection were used. All experiments were evaluated below four passages after transduction. Lentiviral transduction into mpkCCD cells was performed repeatedly to obtain experimental data.

**Cell surface biotinylation assay**. The amount of AQP2 in the apical plasma membrane was quantitated by apical surface biotinylation as previously described[43].

**cAMP assay**. Intracellular cAMP levels were measured with the BioTrak EIA system (GE Healthcare Life Science) as previously described[44]. dDAVP was used as the positive control. Phosphodiesterase inhibitors were not added to the mpkCCD cells.

**Isolated tubule microperfusion experiments**. CCD tubules dissected from 10-week-old female C57BL/6 mice were microperfused in vitro. The dissection, perfusion, and bathing solutions were prepared as previously described[45]. The CCD tubules were dissected in cooled (10 °C) dissection solution, then transferred to a 1.5-mL bathing chamber and microperfused for 1 h before measurements of $P_f$. The perfusion and bathing solutions were bubbled with 95% $O_2$/5% $CO_2$ at room temperature. The perfusion solution contained a volume marker, 0.5 mM fluorescein isothiocyanate-dextran (10,000 MW, Molecular Probes) as previously described[46]. The bathing solution was exchanged every 30 min to maintain osmolality. The perfusion rates were 5–10 nL/min. The tubule lengths were 427 ± 55 μm ($n = 5$). $P_f$ was calculated as previously described[45].

**Synthesis of FMP-API-1/27 (4-benzylchatechol)**. Aluminum chloride (707 mg, 5.3 mmol) was added to a solution of 1,2-dimethoxybenzene (553 mg, 4.0 mmol) and benzoyl chloride (560 mg, 4.0 mmol) in methylene chloride at 0 °C, stirred at room temperature for 1 h, then poured into ice-cooled concentrated hydrochrolic acid and extracted with methylene chloride. The organic layer was dried over sodium sulfate and concentrated. The residue was washed with *n*-hexane to yield 3,4-dimethoxybenzophenone as colorless solid (707 mg, 73%). The parameters of proton nuclear magnetic resonance ($^1$H NMR) (500 MHz, CDCl$_3$) were d 7.77 (dd,

$J = 8.0, 1.5$ Hz, 2 H), 7.59 (tt, $J = 7.2, 1.5$ Hz, 1 H), 7.51 (d, $J = 2.0$ Hz, 1 H), 7.49 (t, $J = 7.8$ Hz, 2 H), 7.39 (dd, $J = 8.5, 2.0$ Hz, 1 H), 6.91 (d, $J = 8.5$ Hz, 1 H), 3.98 (s, 3 H), and 3.96 (s, 3 H).

A solution of 3,4-dimethoxybenzophenone (222 mg, 0.92 mmol) and triethylsilane (836 mg, 7.2 mmol, in methylene chloride (3 mL) and trifluoroacetic acid (2 mL) was stirred at room temperature for 17 h, then poured into ice-cooled saturated sodium bisulfate and extracted with methylene chloride. The organic layer was dried over sodium sulfate and concentrated. The residue was purified by silica gel column chromatography (eluent: $n$-hexane/ethyl acetate = 2/1) to yield 4-benzyl-1,2-dimethoxybenzene as colorless oil (176 mg, 84%). The $^1$H NMR parameters (500 MHz, CDCl$_3$) were d 7.77 (t, $J = 7.2$ Hz, 2 H), 7.23–7.19 (m, 3 H), 6.82 (d, $J = 8.0$ Hz, 1 H), 6.75 (dd, $J = 8.0, 2.0$ Hz, 1 H), 6.73 (d, $J = 2.0$ Hz, 1 H), 4.06 (s, 2 H), 3.87 (s, 3 H), and 3.84 (s, 3 H). Then, 2.0 mL of 1.0 M boron tribromide in methylene chloride was added dropwise to a solution of 4-benzyl-1,2-dimethoxybenzene (176 mg, 0.77 mmol) in methylene chloride at 0 °C. After stirring at room temperature for 2 h, the reaction mixture was poured into ice-cooled water, and extracted with ethyl acetate. The organic layer was dried over sodium sulfate and concentrated to yield 4-benzylchatechol as pale yellow solid (143 mg, 93%). The $^1$H NMR parameters (500 MHz, CDCl$_3$) were d 7.29 (t, $J = 7.5$, 2 H), 7.22–7.17 (m, 3 H), 6.79 (d, $J = 8.0$ Hz, 1 H), 6.69 (d, $J = 2.0$ Hz, 1 H), 6.65 (dd, $J = 8.0, 2.0$ Hz, 1 H), 5.01 (s, 1 H), 4.94 (s, 1 H), and 3.88 (s, 2 H), while those of $^{13}$C NMR (125 MHz, CDCl$_3$) were d 143.6, 141.8, 141.4, 134.6, 129.0, 128.6, 126.2, 121.6, 116.2, 115.5, and 41.3.

**Synthesis of FMP-API-1/28 (3,4,3′,4′-tetrahydroxybenzophenone)**. A mixture of 3,4-dimethoxybenzoic acid (1.5 g, 8.2 mmol) and 1,2-dimethoxybenzene (1.07 g, 7.8 mmol) in 5 g of polyphosphoric acid was stirred by a mechanical stirrer at 80 °C for 2 h[47]. After the mixture was cooled to 60 °C, water was added and the mixture was stirred for an additional 30 min. Colorless precipitates were collected, dissolved in methylene chloride, and washed with 2 M sodium hydroxide and brine. The organic layer was dried over sodium sulfate, and concentrated to yield 3,4,3′,4′-tetramethoxybenzophenone as colorless solid (2.1 g, 84%). The $^1$H NMR parameters (500 MHz, CDCl$_3$) were $d$ 7.44 ($d$, $J = 2.0$ Hz, 2 H), 7.39 (dd, $J = 8.3, 2.0$ Hz, 2 H), 6.92 ($d$, $J = 8.4$ Hz, 2 H), 3.98 (s, 6 H), and 3.95 (s, 6 H). Then, 1.5 mL of 1.0 M boron tribromide in methylene chloride was added dropwise to the solution of 3,4,3′,4′-tetramethoxybenzophenone (101 mg, 0.33 mmol) in methylene chloride at 0 °C. After stirring at room temperature for 3 h, the reaction mixture was poured into ice-cooled water and extracted with ethyl acetate. The organic layer was dried over sodium sulfate and concentrated. The residue was purified by silica gel column chromatography (eluent: CHCl$_3$/MeOH = 15/1) to yield 3,4,3′,4′-tetra-hydroxybenzophenone as colorless solid (57 mg, 70%). The $^1$H NMR parameters (500 MHz, DMSO-$d_6$) were d 9.65 (bs, 2 H), d 9.39 (bs, 2 H), 7.16 (d, $J = 2.0$ Hz, 2 H), 7.05 (dd, $J = 8.2, 2.0$ Hz, 2 H), 6.82 (d, $J = 8.2$ Hz, 2 H), while those of $^{13}$C NMR (125 MHz, DMSO-$d_6$) were d 193.2, 149.7, 144.9, 129.3, 122.7, 116.9, and 114.9.

**Statistical analysis**. Statistical significance was evaluated by a one-way analysis of variance test with multiple comparisons using Tukey's correction. Data are presented as the mean ± standard deviation (SD). The student's $t$-test was performed to assess the statistical significance of microperfusion analysis. Data are presented as the mean ± standard error (SE). A probability ($p$) value of <0.05 was considered statistically significant.

**Data availability**. The data that support the findings of this study are available from the corresponding author upon reasonable request.

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

## Acknowledgements

This work was supported by Grants-in-Aid for Scientific Research (S) (25221306-00 to S.U.), Scientific Research (B) (16H05314 to E.S.), and Research Activity start-up (17H06656 to F.A.) from the Japan Society for the Promotion of Science, a Health Labour Science Research Grant from the Ministry of Health Labour and Welfare, Challenging Exploratory Research from the Ministry of Education, Culture, Sports, Science, and Technology of Japan to S.U. and E.S., the Salt Science Research Foundation (1422 and 1629) to S.U., the Takeda Science Foundation to E.S., Banyu Foundation Research Grant to E.S., the Vehicle Racing Commemorative Foundation to S.U., and TMDU President's Young Researchers Award to F.A.. This work was also partially supported by JSPS Core-to-Core Program A, Advanced Research Networks to H.K.

## Author contributions

F.A. designed, performed, analyzed, and interpreted the experiments and wrote the paper. S.M. and H.K. synthesized the compounds used in the experiments. N.Y., N.N., E.S., T.R., and S.S. analyzed and interpreted the data. T.M. and Y.K. performed, analyzed, and interpreted the microperfusion experiments. S.U. supervised the project, designed and interpreted all the experiments, and wrote the paper.

## Additional information

**Competing interests:** The authors declare no competing interests.

