## [Peer Review File · Nature Communications]

Reviewers' comments:

Reviewer #1 (Remarks to the Author):

This is an interesting article that deals with the role of anchored PKA and modulation of aquaporin water channels. This is a well-studied system and several anchoring proteins known as AKAPs that target PKA to the basolateral membranes of kidney cells have been implicated in this process. In a well-conducted and well-controlled study the authors utilize a variety of well characterized PKA anchoring disruptor peptides to propose that miss-localization of PKA from proximity to the water channel enhances its phosphorylation and activity. They go on to propose that uncoupling of the PKA/AKAP protein-protein interaction activates the kinase and that using a small molecule that targets this interface may have some therapeutic benefits for the treatment of congenital nephrogenic diabetes insipidus. While this article has its merits there are several issues that need to be addressed before it can be considered for publication.

1) The data in figures 1a-d are compelling and interesting. They suggest that miss localization of intact PKA holoenzymes by Ht31 enhance the phosphorylation of cytoplasmic substrates. This is an important observation that augments a recent article in Science showing that anchored PKA holoenzymes remain active and intact. This earlier paper should be cited as it supports the authors conclusion, as should the original references marking the discovery of Ht31 peptide and its proline derivative.

2) The subcellular fractionation data in figure 1e is weak. There would be more confidence in this data if some immunofluorescence images were included showing that the subcellular distribution of PKA was altered upon treatment with Ht31. Also the authors could look to see if they detect enhanced phosphorylation of basolateral substrates with their phosphor-PKA substrates antibody.

3) The data in figure 2 are interesting but counterintuitive to conventional thinking. Ht31 and all of its derivatives displace PKA from AKAPs yet the authors show in fig 2a that phosphorylation of Ser 261 in the cytoplasmic tail of the AQP2 is reduced. This could be explained by enhanced activity of local phosphatase activity. The authors fail to note that, one of the AQP2 associated anchoring proteins, AKAP220 binds PP1. This could provide a mechanistic explanation for this finding. Also, the author should consider that phosphorylation of Ser 269 may be performed by another H89 sensitive kinase.

4) In the remainder of figure 2, the authors conclusively show the involvement of AKAPs in the control of AQP2 regulation. This has been known for some time from the work of Klussmann and colleagues and others. More importantly, does siRNA knockdown of AKAP18, AKAP220 or AKAP79 impact the accumulation of AQP2 at basolateral membranes?

5) The remainder of the article uses the small molecule FMP-API-1 as an antagonist of AKAP-PKA interactions. Another property of this compound is the ability to bind to an allosteric site of regulatory subunits of PKA to activate the intact PKA holoenzyme. This finding is reminiscent of the use of Rp-cAMPS analogs that perform the same function. The authors could bolster their findings by including experiments that utilize these cell soluble compounds. For example, (Sp)-cAMPS sustains active PKA holoenzyme conformations whereas its diastereomer (Rp)-cAMPS constricts and inhibits PKA holoenzymes.

Reviewer #2 (Remarks to the Author):

In this study, Ando and colleagues identify, for the first time, FMP-API-1 as a lead compound for the treatment of congenital nephrogenic diabetes insipidus (NDI) secondary to loss of function of the vasopressin V2 receptor (AVPR2).

The authors worked on the dissociation of A kinase anchoring proteins (AKAPs) binding to PKA to strongly activate AQP2.

They first used the Ht31 inhibitor peptide, a stearylated form of a 24-amino acid peptide derived from a human thyroid AKAP. This peptide competes with AKAPs for PKA RII binding and inhibits interactions between AKAPs and PKA RII. As demonstrated in Fig. 1b, Ht31 and dDAVP, but not Ht31P, significantly increased PKA activity. Ht31 phosphorylated AQP2 at S269 and dephosphorylated AQP2 at S261, similar to dDAVP (fig 2A C).

Apical surface biotinylation showed that Ht31 increased apical AQP2 expression (Fig. 2d). These results indicated that the Ht31-induced PKA activation stimulated AQP2 trafficking and apical AQP2 accumulation.

.

Since the in vivo efficacy of Ht31 is limited because Ht31 is a peptide drug that is characterized by a short half-life and low membrane permeability the authors then used another AKAPs-PKA disruptor, FMP-API-1, which is a low molecular weight compound with enhanced plasma membrane permeability. The effects of FMP-API-1 in mpkCCD cells were extremely similar to those of Ht31. FMP-API-1 increased AQP2 activity and water permeability in vitro and the efficacy of FMP-API-1 in AQP2 trafficking was almost equal to that of dDAVP. Functional analysis of osmotic water permeability (Pf) in isolated tubules of the CCDs of mouse kidneys was examined using a microperfusion technique. As shown in Fig. 4e, FMP-API-1 significantly increased Pf, as compared to the control values. The effect of FMP-API-1 was equivalent to that of dDAVP.

FMP-API-1 increases urine concentration in an NDI mouse model. Tolvaptan, a vasopressin V2 receptor antagonist, decreased urine osmolality and increased urine output and water intake. Administration of FMP-API-1 significantly attenuated the effects of tolvaptan.

The derivatives of FMP-API-1 robustly activates AQP2. FMP-API-1/27-phosphorylation of AQP2 was tissue-specific providing hope that the clinical use of this compound will also be tissue specific and will increase urinary osmolality in NDI patients without interfering with other signaling pathways.

Critiques : the data presented here are new and important clinically. Fig 6 could also include some elements of figure 1 of a recent review (Cell Signal. 2017 Sep;37:1-11. doi:

10.1016/j.cellsig.2017.05.012. Epub 2017 May18. The many faces of compartmentalized PKA signalosomes.)

The ko model of NDI (J Clin Invest. 2009 Oct;119(10):3115-26. A selective EP4 PGE2 receptor agonist alleviates disease in a new mouse model of X-linked nephrogenic diabetes insipidus.

Li JH1, Chou CL, Li B, Gavrilova O, Eisner C, Schnermann J, Anderson SA, Deng CX, Knepper MA, Wess J.) with conditionally deletion of the Avpr2 gene during adulthood by administration of 4-OH-tamoxifen could have been used but the tolvaptan data presented here are clear and the next step to me will be, after appropriate safety studies, testing humans.

Point-by-point reply to reviewers

We thank the reviewers for their thoughtful comments, which have helped to improve the manuscript. We feel that we could successfully address most of the previously raised concerns. Our replies to the reviewer's comments are below.

Reviewer #1 (Remarks to the Author):

This is an interesting article that deals with the role of anchored PKA and modulation of aquaporin water channels. This is a well-studied system and several anchoring proteins known as AKAPs that target PKA to the basolateral membranes of kidney cells have been implicated in this process. In a well-conducted and well-controlled study the authors utilize a variety of well characterized PKA anchoring disruptor peptides to propose that miss-localization of PKA from proximity to the water channel enhances its phosphorylation and activity. They go on to propose that uncoupling of the PKA/AKAP protein-protein interaction activates the kinase and that using a small molecule that targets this interface may have some therapeutic benefits for the treatment of congenital nephrogenic diabetes insipidus. While this article has its merits there are several issues that need to be addressed before it can be considered for publication.

1) The data in figures 1a-d are compelling and interesting. They suggest that miss localization of intact PKA holoenzymes by Ht31 enhance the phosphorylation of cytoplasmic substrates. This is an important observation that augments a recent article in Science showing that anchored PKA holoenzymes remain active and intact. This earlier paper should be cited as it supports the authors conclusion, as should the original references marking the discovery of Ht31 peptide and its proline derivative.

Response:

We thank the reviewer for the important points.

As the reviewer pointed out, a recent article in Science strongly supports our results.

We cited the article below in **p.9**.

- Science. 2017 June 23; 356(6344): 1288–1293.

In addition, we cited the articles below about Ht31 in **p.8**.

- J Biol Chem. 1992 Jul 5;267(19):13376-82.
- J Biol Chem. 1997 Feb 21;272(8):4747-52.

2-1) The subcellular fractionation data in figure 1e is weak. There would be more confidence in this data if some immunofluorescence images were included showing that the subcellular distribution of PKA was altered upon treatment with Ht31.

Response:

We would like to thank for thoughtful reviewer`s comments.

In response to reviewer`s comments, we examined the subcellular distribution of PKA RII α / β using immunofluorescent staining and confocal microscopy. Although we tried the several commercially available antibodies to PKA RII α / β , the antibodies we used did not detect signals of PKA RII α / β .

We then generated stable cell lines that express PKA RII α -GFP or PKA RII β -GFP to examine the localization of PKA. However, such overexpression of PKA RII α -GFP or PKA RII β -GFP inhibited the Ht31-induced phosphorylation of PKA substrates (Fig. 1b) and AQP2 (Fig. 2a), as shown in figure below (**Figure a**). Also, the subcellular distribution of PKA RII α -GFP or PKA RII β -GFP in these stable cell lines was not changed by Ht31 (**Figure b**). Ht31 might be mainly trapped to the overexpressed PKA RII α / β -GFP, and could not exert the function as a AKAPs-PKA disruptor. Thus, we could not evaluate the subcellular localization of PKA RII α / β -GFP in the mpkCCD cells in this revision, but this result in the overexpression of PKA RII α -GFP or PKA RII β -GFP would rather ensure our data that the effects of Ht31 on PKA and AQP2 were mediated by PKA RII α / β .

We added “We further examined the subcellular distribution of PKA RII α / β using immunofluorescent staining; however, the antibodies we used did not detect signals of PKA RII α / β .” in **p.9**.

In addition, we added “In addition, these changes in phosphorylation status (Fig. 1b,2a) were also inhibited by the overexpression of PKA RII α / β (Supplementary Fig. 4). Ht31 might be mainly trapped to the overexpressed PKA RII α / β , and could not exert the function as a AKAPs-PKA disruptor. This result ensured that the effects of Ht31 on PKA and AQP2 were mediated by PKA RII α / β .” in **p.10**.

2-2) Also the authors could look to see if they detect enhanced phosphorylation of basolateral substrates with their phosphor-PKA substrates antibody.

Response:

As the reviewer pointed out, PKA activity in membrane fraction was increased after PKA was translocated from cytosol to membrane fraction, as shown in figure below.

We added “Importantly, PKA activity in membrane fraction was increased after PKA was translocated from cytosol to membrane fraction (Fig. 1f).” in **p.9**.

3) The data in figure 2 are interesting but counterintuitive to conventional thinking. Ht31 and all of its derivatives displace PKA from AKAPs yet the authors show in fig 2a that phosphorylation of Ser 261 in the cytoplasmic tail of the AQP2 is reduced. This could be explained by enhanced activity of local phosphatase activity. The authors fail to note that, one of the AQP2 associated anchoring proteins, AKAP220 binds PP1. This could provide a mechanistic explanation for this finding. Also, the author should consider that phosphorylation of Ser 269 may be performed by another H89 sensitive kinase.

Response:

We thank the reviewer’s important point. AQP2 phosphorylation at S261 and S269 is the most important factors/indicators for AQP2 trafficking. Thus, many AQP2 researchers have examined which kinases and phosphatases are responsible for the AQP2 phosphorylation; however, the responsible molecules and exact mechanisms are currently not fully understood.

Although PP1 is a one of the potential regulators for AQP2 phosphorylation at S261 through interaction with AKAP220, it seems difficult to clarify the involvement of PP1 in AQP2 phosphorylation because of the following reasons.

1.

So far, PP1 is not considered to be responsible for AQP2 de-phosphorylation at S261. PP1 and PP2A inhibitor calyculin did not change AQP2 phosphorylation at S261 and S269 in the analysis of *ex vivo* kidney tissue (*Am J Physiol Renal Physiol.* 2016; 311(6): F1189-F1197.). In addition, PP1 and PP2A inhibitor okadaic acid did not affect AQP2 phosphorylation at S261 in LLC-AQP2 cells (*Am J Physiol Renal Physiol.* 2017; 313(2): F404-F413.).

2.

Mitogen-activated protein kinases (MAPKs) are already known to directly phosphorylate AQP2 peptide at S261 *in vitro* (*Proc Natl Acad Sci U S A.* 2010; 107(8): 3882–3887.). In mpkCCD cells, the activities of MAPKs are reduced by PKA, suggesting that AQP2 de-phosphorylation at S261 is regulated by PKA/MAPKs signaling (*Proc Natl Acad Sci U S A.* 2017; 114(42): E8875-E8884.).

In fact, PKA activators, such as Ht31 and dDAVP, clearly decreased phosphorylation and activities of MAPKs (p-p38, pJNK, pERK) in mpkCCD cells, as shown in figure below.

Therefore, in the regulation of AQP2 phosphorylation at S261, several molecules other than PP1 would also be simultaneously modulated by PKA.

3.

We further examined the effects of PP1 and PP2A inhibitors okadaic acid (OA) (100 nM) and tautomycetin (TMC) (1 μ M) in mpkCCD cells, as shown in figure below.

TMC did not inhibit VP-induced de-phosphorylation of AQP2 at S261 and phosphorylation of AQP2 at S269 **(a)**. On the other hand, OA inhibited the effects of dDAVP on AQP2 phosphorylation **(a)**; however, immunofluorescence study showed that OA also induced cell shrinkage and the loss of cell polarity **(b)**.

Inhibition of PP1 probably exerted toxic effects on mpkCCD cells, resulting in the alteration of AQP2 phosphorylation at S261 and S269. Pharmacological inhibitors of PP1 were not specific to AKAPs-binding PP1 that regulates AQP2 activity.

4.

Pharmacological inhibition experiments are not enough to evaluate the precise role of AKAPs-binding phosphatases on AQP2. Knockin cell lines that express AKAPs mutants deleting the binding domain of phosphatases are required (*Cell Rep.* 2014; 7(5): 1577–1588.).

In our future studies, we would like to examine the effects of AKAPs-binding phosphatases on AQP2 phosphorylation using these experimental methods.

Although the precise mechanisms are still unknown, Ht31 probably affects local activities of kinases and phosphatases responsible for AQP2 phosphorylation at S261 and S269 through PKA activation. Therefore, we changed schematic summary in Figure 6 and referred to these points.

4) In the remainder of figure 2, the authors conclusively show the involvement of AKAPs in the control of AQP2 regulation. This has been known for some time from the work of Klusmann and colleagues and others. More importantly, does siRNA knockdown of AKAP18, AKAP220 or AKAP79 impact the accumulation of AQP2 at basolateral membranes?

Response:

We examined the expression of representative endogenous AKAPs in mpkCCD cells by RT-PCR, as shown in figure below.

N.C. indicates negative control (water in place of cDNA); cDNA from heart, testis, kidney, and brain of C57BL/6 mouse are used as positive control.

In the mpkCCD cells, more than 19 AKAPs including *Akap5* (AKAP79/150), *Akap7* (AKAP18), *Akap11* (AKAP220) are expressed. Knockdown of a sole AKAP may be insufficient to impact the accumulation of AQP2 at plasma membranes because Ht31 and FMP-API-1 dissociated various combinations and interactions of AKAPs and PKA.

Analyses of knockout mice also support this notion. Although the role of AKAP18 and AKAP220 in the regulation of AQP2 have been elucidated *in vitro*, both AKAP18 and AKAP220 knockout mice did not present a phenotype of AQP2 activation, such as the syndrome of inappropriate antidiuretic hormone secretion, in the steady-state condition (*Proc Natl Acad Sci U S A.* 2012; 109: 17099-104, *Proc Natl Acad Sci U S A.* 2016; 113: E4328-37.). AQP2 accumulation at plasma membranes in AKAP220 knockout mice was observed only in overhydrated states. These results suggest that simultaneous knockdown or knockout of several AKAPs may be required to activate AQP2 sufficiently. Nonetheless, as the reviewer suggested, the identification of major AKAPs involved in

AQP2 regulation by Ht31 and FMP-API-1 is important. We would like to focus on this issue in our future studies.

We added the figure above in Supplementary Fig.1 to indicate potential targets of Ht31 and FMP-API-1.

5) The remainder of the article uses the small molecule FMP-API-1 as an antagonist of AKAP-PKA interactions. Another property of this compound is the ability to bind to an allosteric site of regulatory subunits of PKA to activate the intact PKA holoenzyme. This finding is reminiscent of the use of Rp-cAMPS analogs that perform the same function. The authors could bolster their findings by including experiments that utilize these cell soluble compounds. For example, (Sp)-cAMPS sustains active PKA holoenzyme conformations whereas its diastereomer (Rp)-cAMPS constains and inhibits PKA holoenzymes.

Response:

We would like to thank for reviewers' valuable suggestions. We examined the effects of Sp-cAMPS and Rp-cAMPS in mpkCCD cell lines.

In previous reports, Sp-cAMPS and Rp-cAMPS were used at concentrations of approximately 50-100 μ M. Similarly, Sp-cAMPS at concentrations above 50 μ M clearly increased phosphorylation of PKA substrates and AQP2 in mpkCCD cells, as shown in figure below. The effects of Sp-cAMPS at 150 μ M was almost equal to those of vasopressin (dDAVP).

In contrast, Rp-cAMPS at concentrations above 50 μ M did not increase phosphorylation of PKA substrates and AQP2, as shown in figure below. In addition, Rp-cAMPS at 150 μ M did not increase the effects of FMP-API-1 on phosphorylation of PKA substrates and AQP2.

For this reason, Rp-cAMPS (150 μ M) did not dissociate the binding of AKAPs and PKA, as shown in figure below. In contrast to FMP-API-1, Rp-cAMPS did not increase PKA RII α / β protein expression in the membrane fraction (MF). Although both Rp-cAMPS and FMP-API-1 bind to an allosteric site of regulatory subunits of PKA, only FMP-API-1 changed the subcellular localization of PKA RII α / β and activated PKA and AQP2.

(Similar to vasopressin (dDAVP), Sp-cAMPS (150 μ M) increased the lower band of PKA RII α in MF.)

Reviewer #2 (Remarks to the Author):

In this study, Ando and colleagues identify, for the first time, FMP-API-1 as a lead compound for the treatment of congenital nephrogenic diabetes insipidus (NDI) secondary to loss of function of the vasopressin V2 receptor (AVPR2).

The authors worked on the dissociation of A kinase anchoring proteins (AKAPs) binding to PKA to strongly activate AQP2.

They first used the Ht31 inhibitor peptide, a steared form of a 24-amino acid peptide derived from a human thyroid AKAP. This peptide competes with AKAPs for PKA RII binding and inhibits interactions between AKAPs and PKA RII. As demonstrated in Fig. 1b, Ht31 and dDAVP, but not Ht31P, significantly increased PKA activity. Ht31 phosphorylated AQP2 at S269 and dephosphorylated AQP2 at S261, similar to dDAVP (fig 2A C).

Apical surface biotinylation showed that Ht31 increased apical AQP2 expression (Fig. 2d). These results indicated that the Ht31-induced PKA activation stimulated AQP2 trafficking and apical AQP2 accumulation.

Since the in vivo efficacy of Ht31 is limited because Ht31 is a peptide drug that is characterized by a short half-life and low membrane permeability the authors then used another AKAPs-PKA disruptor, FMP-API-1, which is a low molecular weight compound with enhanced plasma membrane permeability. The effects of FMP-API-1 in mpkCCD cells were extremely similar to those of Ht31. FMP-API-1 increased AQP2 activity and water permeability in vitro and the efficacy of FMP-API-1 in AQP2 trafficking was almost equal to that of dDAVP. Functional analysis of osmotic water permeability (Pf) in isolated tubules of the CCDs of mouse kidneys was examined using a microperfusion technique. As shown in Fig. 4e, FMP-API-1 significantly increased Pf, as compared to the control values. The effect of FMP-API-1 was equivalent to that of dDAVP.

FMP-API-1 increases urine concentration in an NDI mouse model. Tolvaptan, a vasopressin V2 receptor antagonist, decreased urine osmolality and increased urine output and water intake. Administration of FMP-API-1 significantly attenuated the effects of tolvaptan.

The derivatives of FMP-API-1 robustly activates AQP2. FMP-API-1/27-phosphorylation of AQP2 was tissue-specific providing hope that the clinical use of this compound will also be tissue specific and will increase urinary osmolality in NDI patients without interfering with other signaling pathways.

Critiques :

1) the data presented here are new and important clinically. Fig 6 could also include some elements of figure 1 of a recent review (Cell Signal. 2017 Sep;37:1-11. doi: 10.1016/j.cellsig.2017.05.012. Epub 2017 May18.The many faces of compartmentalized PKA signalosomes.)

Response:

We thank the reviewer's important points. According to reviewer's comments, we revised original Figure 6, as shown in figure below.

We added "AKAPs generally contain protein-protein interactions (PPIs) sites, amphipathic helix domain, and targeting domain. The amphipathic α -helix enables the anchoring of PKA holoenzymes. The targeting domain mediates the tethering of PKA complexes to selected subcellular compartments. FMP-API-1 and -1/27 dissociated PKA from AKAPs. The changes in PKA localization increased PKA activity in renal collecting ducts. PKA as well as additional kinases and phosphatases which are interacted with AKAPs through PPIs sites coordinately regulated AQP2 phosphorylation, trafficking, and water transport." in **legend for Figure 6**.

In addition, we cited the articles above.

2) The ko model of NDI (J Clin Invest. 2009 Oct;119(10):3115-26. A selective EP4 PGE2 receptor agonist alleviates disease in a new mouse model of X-linked nephrogenic diabetes insipidus.

Li JH1, Chou CL, Li B, Gavrilova O, Eisner C, Schnermann J, Anderson SA, Deng CX, Knepper MA, Wess J.) with conditionally deletion of the Avpr2 gene during adulthood by administration of 4-OH-tamoxifen could have been used but the tolvaptan data presented here are clear and the next step to me will be, after appropriate safety studies, testing humans.

Response:

Thank you for reviewer's encouraging comments. In our future studies, we further investigate potential risks and side effects of FMP-API-1 for future human studies.

REVIEWERS' COMMENTS:

Reviewer #1 (Remarks to the Author):

The authors have done an excellent job responding to the reviewers comments. I am ready to accept this article. One word of caution, there is still considerable controversy and some skepticism about the mechanism of action of the small molecule "anchoring inhibitor compound" that the authors are using. It might be a prudent move to add a sentence or two about the limitations of this reagent as many investigators have abandoned the use of this compound.

Point-by-point reply to reviewers

We thank the reviewers for their thoughtful comments, which have helped to improve the manuscript. Our replies to the reviewer's comments are below.

Reviewers' comments:

Reviewer #1

The authors have done an excellent job responding to the reviewers comments. I am ready to accept this article. One word of caution, there is still considerable controversy and some skepticism about the mechanism of action of the small molecule "anchoring inhibitor compound that the authors are using. It might be a prudent move to add a sentence of two about the limitations of this reagent as many investigators have abandoned the use of this compound.

Response:

We would like to thank for thoughtful reviewer`s comments.

We added "Whereas, there is a limitation to develop clinically applicable AKAPs-PKA disruptors. The precise mechanisms underlying the effectiveness of small molecule inhibitors of protein-protein interactions, such as FMP-API-1/27, for clinical use are still controversial. Further studies are required to determine intracellular target molecules of these drug candidates." in p.18.